# Cutaneous Lymphoma and Antibody-Directed Therapies

**DOI:** 10.3390/antib12010021

**Published:** 2023-03-03

**Authors:** Alvise Sernicola, Christian Ciolfi, Paola Miceli, Mauro Alaibac

**Affiliations:** Dermatology Unit, Department of Medicine (DIMED), University of Padua, 35121 Padova, Italy

**Keywords:** cutaneous lymphoma, monoclonal antibodies, rituximab, brentuximab vedotin, photoimmunotherapy, antibody–drug conjugates, bi-specific T cell engager, radioimmunotherapy, antibody combinations, nanobioconjugates

## Abstract

The introduction of monoclonal antibodies such as rituximab to the treatment of cancer has greatly advanced the treatment scenario in onco-hematology. However, the response to these agents may be limited by insufficient efficacy or resistance. Antibody–drug conjugates are an attractive strategy to deliver payloads of toxicity or radiation with high selectivity toward malignant targets and limited unwanted effects. Primary cutaneous lymphomas are a heterogeneous group of disorders and a current area of unmet need in dermato-oncology due to the limited options available for advanced cases. This review briefly summarizes our current understanding of T and B cell lymphomagenesis, with a focus on recognized molecular alterations that may provide investigative therapeutic targets. The authors reviewed antibody-directed therapies investigated in the setting of lymphoma: this term includes a broad spectrum of approaches, from antibody–drug conjugates such as brentuximab vedotin, to bi-specific antibodies, antibody combinations, antibody-conjugated nanotherapeutics, radioimmunotherapy and, finally, photoimmunotherapy with specific antibody–photoadsorber conjugates, as an attractive strategy in development for the future management of cutaneous lymphoma.

## 1. Introduction

In recent decades, the introduction of monoclonal antibodies (mAbs) has revolutionized the therapeutical approach to cancer. At the beginning of the 20th century, Paul Erlich first introduced the idea of “magic bullets”, proposing the development of new compounds targeting specific tumor surface antigens, without any harmful effect on normal cells [1]. The first mAb to be approved by the US Food and Drug Administration (FDA) in 1997 was rituximab for the treatment of relapsed or refractory, CD20-positive, B cell, low-grade or follicular non-Hodgkin’s lymphoma (NHL) [2]. Since that moment, an increasing number of mAbs have been developed for cancer treatment as a valid alternative to traditional cytotoxic agent-based chemotherapy, which is penalized by low tumor selectivity with consequent unwanted and harmful off-target effects. However, mAbs alone do not often provide sufficient cytotoxic effects, resulting in limited response when employed as monotherapy [3]. This holds true for the mAbs currently available in the setting of cutaneous lymphoma which achieve response in around half of patients, with ample variations according to stage and differential involvement of the skin and blood compartments [4,5].

To overcome these limitations, antibody–drug conjugates (ADCs) have emerged as a new strategy to combine the high cytotoxicity of traditional chemotherapy with the selectivity of mAbs in the treatment landscape of targeted therapies [6]. These compounds consist of three main elements: a mAb against a target antigen, a cytotoxic agent, and a chemical linker. The ideal target antigen should be expressed highly and homogenously on cancer cells and minimally on healthy tissues, to limit off-tumor toxicity. Proteins expressed on tumor cell surfaces constitute preferential targets; however, intracellular antigens may also be employed [7]. As for cytotoxic agents, the most commonly used are microtubule inhibitors (such as auristatins and maytansine derivatives), DNA-damaging agents (such as calicheamicins, and pyrrolobenzodiazepines dimers) and topoisomerase inhibitors. MAbs were first developed in murine models; in order to reduce their immunogenicity when used in humans, chimeric antibodies and humanized antibodies were developed, by combining sequences of the murine variable domain with human constant domain and by inserting the murine hypervariable regions into a human antibody, respectively. Finally, the development of additional techniques allowed fully human monoclonal antibodies to be derived [6,7]. MAbs used to develop ADCs usually belong to the IgG subtypes and are humanized or chimeric to minimize their immunogenicity; they are linked to cytotoxic agents through a chemical linker, which can be cleavable or non-cleavable [6,8,9]. The pharmacokinetics of mAbs mainly involves non-linear and saturable clearance, which is mediated by binding to the target, a cell surface receptor of peripheral blood mononuclear cells. Additionally, at higher doses, non-saturable proteolytic pathways are able to degrade IgG antibodies similarly to endogenous IgG [6]. Finally, it should be considered that cytotoxic agents may also exhibit activity on adjacent cells that are target negative: this may lead either to further tumor cell or tumor microenvironment killing (the so-called bystander effect) or to off-target cytotoxicity if normal cells are involved [10,11].

An additional advancement in the field of antibody-directed therapies consists of the development of antibody-functionalized nanotherapeutics. The generation of nanobioconjugates, that is drug-containing nanocarriers, is an attractive option for selective cancer therapy [12,13]. A further level of selectivity is provided by making these nanostructures responsive to specific stimuli, including externally applied irradiation [14,15]. Antibody-conjugated nanomedicine could combine the target-specific binding of mAbs with enhanced delivery and on-demand release of drugs, achieving high efficacy and excellent safety in future oncologic therapeutics.

Primary cutaneous lymphomas constitute a heterogeneous group of disorders characterized by monoclonal proliferations of lymphocytes with infiltration primarily involving the skin, modified skin appendages and certain mucosal sites (Table 1). Several treatment options are available for primary cutaneous lymphomas; however, therapy for advanced cases remains largely palliative, with allogenic hematopoietic stem cell transplant representing the only potentially curative option. While regimens incorporating anti-CD20 mAbs may be considered the current standard of care in cutaneous B cell lymphoma (CBCL), the optimal management for advanced cutaneous T cell lymphoma (CTCL) has not been defined [16]. A sequential multidisciplinary approach based on the disease stage and tailored individually is the preferred strategy and incorporates the sequential use or combination of biologic-response-modifying drugs [17], histone deacetylase inhibitors [18,19], and extracorporeal photopheresis [20]. Moreover, escalated chemotherapy regimens may provide elevated rates of response but are burdened by adverse events—namely, infections and myelosuppression—and by a short duration of responses [21,22]. For these reasons, traditional chemotherapy is usually considered following multiples relapses or for the debulking of extensive visceral disease [21]. Relapses are also frequent following autologous stem cell transplant and only allogenic transplant may provide durable responses, mediated by a graft-versus-lymphoma reaction [23,24]. Finally, targeted agents require robust evidence supporting their use alone or in combination and additional novel approaches are urgently needed [25,26,27]. This review briefly summarizes our current understanding of the pathophysiology of T and B cell lymphomagenesis, with a focus on recognized molecular alterations that may provide investigative therapeutic targets. Then, evidence on the use of antibody-directed therapies in the setting of lymphoma is discussed: this broad definition includes ADCs as well as bi-specific T cell engagers, antibody combinations, antibody-functionalized nanotherapeutics, and radioimmunotherapy. Finally, the development of antibodies conjugated with a photoadsorber molecule is presented as an attractive strategy for the future management of cutaneous lymphoma, providing target-specific binding with the additional advantage of limiting activation to the malignant tissue irradiated by light. This latter approach is particularly suitable in cutaneous oncology considering that the skin compartment is readily accessible to the effect of an external light source.

## 2. Cutaneous T Cell Lymphoma

### 2.1. Definition and Classification

In the Western world, CTCLs represent approximately 75 to 80% of all primary cutaneous lymphomas [28,29]. Specifically, mycosis fungoides (MF), Sezary syndrome (SS), and primary cutaneous CD30+ lymphoproliferative disorders (LPD) account for 55, 5 and 30% of CTCL, respectively.

MF represents almost 55% of cases of CTCL; three peculiar variants with distinctive clinicopathologic features are included within this diagnosis: folliculotropic MF, pagetoid reticulosis, and granulomatous slack skin. SS accounts for only 5% of cases and is a leukemic type of CTCL presenting with erythroderma and peripheral lymphadenopathy. The detection of neoplastic T cells (Sezary cells) in skin, lymph nodes, and peripheral blood is characteristic of SS [30,31].

Primary cutaneous CD30+ LPD is the term for a group of disorders that represent approximately 30% of CTCL overall. This group includes primary cutaneous anaplastic large cell lymphoma (PCALCL), lymphomatoid papulosis (LyP), and borderline cases.

Additional entities are: adult T cell leukemia-lymphoma, a peripheral T cell neoplasm associated with the human T cell leukemia virus 1 (HTLV-1); subcutaneous panniculitis-like T cell lymphoma, which has an alpha-beta T cell phenotype associated with an indolent biologic behavior; and extranodal NK/T cell lymphoma, nasal type, which is in almost all cases an Epstein–Barr virus (EBV)-positive lymphoma composed of small, medium, or large cells (with NK or, rarely, cytotoxic T cell phenotype).

Finally, rare subtypes of primary cutaneous peripheral T cell lymphoma have been recognized, including provisionally classified entities: primary cutaneous gamma-delta T cell lymphoma, primary cutaneous aggressive epidermotropic CD8+ cytotoxic T cell lymphoma, primary cutaneous acral CD8+ T cell lymphoma, primary cutaneous CD4+ small/medium T cell LPD [29].

### 2.2. Pathophysiology of T Cells

Local immune and inflammatory responses in the skin are dependent on the migratory abilities of T cells that are tightly regulated by lymphocyte-endothelial interactions [32]. Homing to the cutaneous compartment is a multistep process. Adhesion between lymphocytes and endothelial cells is mediated by molecules expressed on the cell surface and promoted by the secretion of chemokines which occurs in the context of local inflammation. Extravasation begins when the cutaneous lymphocyte antigen (CLA) on the lymphocyte interacts with E-selectin on activated endothelium; this initial tethering brings the cell close to the stimuli of local chemokines such as CCR4, which activate surface integrins. The latter provide firm anchoring to intercellular adhesion molecules on the endothelium [33]. Immune cells expressing the specific homing receptor CLA include approximately 15% of memory T cells as well as NK cells, monocytes, granulocytes, and a small percentage of B cells [34,35,36]. Owing to the key role of CLA for cutaneous homing, this surface molecule is expressed on malignant cells across a variety of primary cutaneous lymphomas [37].

CTCL are generally derived from memory subsets, mainly CD4+, of T cells that are associated with the expression of CLA together with CCR4 chemokine receptor [38]. This is the typical homing signature of MF cells, accounting for the prolonged localization of MF to specific skin sites [16,39]. Conversely, malignant T cells in SS, a CTCL entity closely related to MF, co-express L-selectin and CCR7, that are the lymph node homing signals of central memory T cells, together with CLA and CCR4 [40]. Additionally, when MF spreads to the lymph nodes CCR7 is expressed and skin homing receptors are downregulated [41]. Apart from MF and SS, CLA has been detected in CD30+ LPDs and in other rare subtypes of CTCL, including primary cutaneous aggressive epidermotropic CD8+ cytotoxic T cell lymphoma [37]. Finally, though its expression has been reported in isolated cases of primary cutaneous B cell lymphoma, the involvement of CLA in the genesis of malignant B cells is likely marginal [37].

### 2.3. Distinctive Molecular Alterations of CTCL

Alternations of cellular signaling pathways and imbalances of the skin immune microenvironment have been identified to be at the basis of CTCL. Chromosome translocations leading to activation of oncogenes, deletions or mutations inactivating tumor suppressor genes, and point mutations affecting the epigenetic control of transcription or translation have been increasingly recognized.

The pathogenesis of malignancy in general is a complex process in which increasing genetic damage affects proto-oncogenes and tumor suppressor genes; lymphomagenesis is no exception to this rule. In many subtypes of lymphoma, the genome is characterized by few non-random balanced chromosomal translocations and additional unbalanced alterations that usually accumulate during disease progression. These lesions may be responsible for the activation of oncogenes, while specific deletions suggest inactivation of tumor suppressor genes. However, the spectrum of somatic mutations associated with CTCL is only partially understood. Genome sequencing approaches have been employed to investigate disease pathways that undergo somatic mutation in CTCL highlighting the role of MAPK, NF-kB, PI3K, and TCR; genes related to functions of immune surveillance and RNA splicing have been additionally implicated [42,43].

Recent investigations have demonstrated the importance of MAPK, PI3K/Akt, JAK/STAT, NF-kB, TCR and TLR downstream signaling to allow survival in CTCL. These studies show that cell signaling is differentially altered in different stages and cell populations of CTCL. Specifically, JAK/STAT signaling dysregulation has been described in both early and advanced disease. In early CTCL, activation of STAT signaling is largely dependent on IL-2, IL-7, IL-15 [44].

Different steps of the TCR signaling pathway, from complex formation to activation of transcription factors, also showed abnormalities in the immune microenvironment of CTCL [45].

As far as MF is concerned, its pathophysiology is still incompletely understood but specific molecular pathways have been implicated: TCR and JAK-STAT signaling, RNA splicing and epigenetic control. An increased MAPK signaling due to gain of function mutations in B-RAF and MAPK3 has been demonstrated in subjects with MF [46]. The key proteins in the PI3K/Akt pathway—namely, Akt, mTOR, p70S6K—have been associated with advanced MF stage [47].

Finally, a correlation of CTCL with environmental and occupational exposure to solvents and chemicals has been hypothesized but robust evidence to support this relationship is lacking. In the case of MF, it has been demonstrated that UV genetic changes are associated with altered cell signaling and changes in the skin microenvironment, with gain of function in proto-oncogenes, loss of function in tumor suppressors and adhesion proteins [48].

The currently available antibody-directed treatments for cutaneous lymphoma are summarized in Figure 1. Novel investigative strategies for drug discovery in CTCL will be provided by targeting the key proteins of signaling pathways that stimulate malignant T cells and firing up antitumor immune responses in the CTCL microenvironment [49].

## 3. Cutaneous B Cell Lymphoma

### 3.1. Definition and Classification

CBCLs are defined as cases of B cell lymphoma that present in the skin without detectable extracutaneous disease. The skin is the second primary extranodal lymphoma site in frequency, following the gastrointestinal tract. Our understanding of cutaneous lymphomas is largely dependent on the development of immunophenotyping and molecular testing techniques, that allowed the classification of current entities. CBCLs constitute up to 25% of all primary cutaneous lymphomas and can be sorted into three main subtypes according to the 2005 WHO-EORTC classification for cutaneous lymphomas [28] and its 2018 revision [29]: (1) primary cutaneous follicle center lymphoma (PCFCL) is the most frequent subtype of primary CBCL, accounts for roughly half the cases, and has a clinically indolent course with involvement limited to the skin; unlike nodal follicle center lymphoma, PCFCL is usually BCL2 negative and lacks the hallmark chromosomal translocation t(14;18); (2) primary cutaneous marginal zone lymphoma (PCMZL) is an indolent entity that makes up 30% of CBCL; two subtypes are recognized according to Ig heavy-chain class: IgG-positive class-switched PCMZL, which is the most common type and shows an indolent course, and IgM-positive non-class-switched PCMZL, which is uncommon and tends to progress to extracutaneous involvement; (3) primary cutaneous diffuse large B cell lymphoma, leg type (PCDLBCL, LT) is an intermediate/aggressive type that accounts for 20% of primary CBCL.

### 3.2. Pathophysiology of B Cells

While the immunosurveillance functions of T lymphocytes require that these cells undergo extensive trafficking in various peripheral non-lymphoid organs, the effector functions of B lymphocytes are largely mediated by soluble antibodies and can be performed at a distance from the target tissue [32]. Contrary to the normal recruitment of skin homing T cells, the cutaneous compartment normally does not contain detectable B cells, since they are not required to migrate to the local sites of antigen stimulation [50]. However, persistent inflammation consequent to chronic infection—caused by Borrelia burgdorferi—and autoimmunity can trigger the development of newformed lymphoid tissue [51,52]; it is in this context that B cell lymphomagenesis is promoted in the skin [53] and may lead to the development of PCMZL and PCFCL. Localized expression of lymphoid tissue-associated chemokines and vascular addressins by activated endothelia supports migration of B cells to the skin with mechanisms that are unrelated to the signals for T cell skin homing that have been discussed above. Moreover, the homing propensity of B cells varies according to their stage of maturation: naïve B cells must recirculate between peripheral and lymphoid tissues to increase chances of antigen encounter, while memory B cells and plasmablasts are directed to lymphoid tissues. A differential surface expression of adhesion molecules and chemokine receptors enables these functions specific to the maturation stage: naïve B cells display L-selectin and integrins alphaLbeta2 and alpha4beta1, which support homing to multiple sites including peripheral lymph nodes and areas of inflammation; memory B cells additionally express chemokine receptors CXCR4 or 5 that are related to the organization of germinal centers, under the guide of ligands CXCL12 or CXCL13, respectively [54,55,56]. These receptors are broadly expressed by B cell lymphoma cells highlighting their key role in both normal B cell homing and malignant lymphoma cell migration.

### 3.3. Distinctive Molecular Alterations of CBCL

To date, scant evidence is available on the molecular alterations associated with the peculiar behavior of indolent CBCL and on the biological differences that set them apart from aggressive forms. Understanding the differences between indolent and aggressive lymphomas with the same cell of origin could help us identify the molecular mechanisms associated with their selective tissue-specific localization or with their systemic dissemination, respectively. This information may ultimately provide potential targets for drug discovery in the setting of lymphoproliferative disorders in the skin and beyond.

Our understanding of primary CBCL is currently incomplete: the implementation of antigen detection techniques based on PCR and FISH to the diagnostic workup of CBCL has provided helpful tools to distinguish indolent forms of CBCL, such as PCFCL, from aggressive subtypes, such as systemic follicle center lymphoma (FCL) with skin involvement and PCDLBCL, LT [28,57].

While PCFCL and PCDLBCL, LT show different clinical and histologic presentations, they may share BCL2 expression, which is a traditional hallmark of systemic FCL: BCL2 is strongly expressed in PCDLBCL, LT and observed mostly in low-grade PCFCL suggesting that this abnormality is not related to increased aggressivity in cutaneous forms [58]. These observations suggest the existence of different unknown genetic abnormalities related to the peculiar behavior of the indolent forms of CBCL, that sets them apart both from aggressive cutaneous subtypes and from their nodal counterparts [59,60].

PCDLBCL, LT is markedly different from PCFCL but has genetic similarities with nodal diffuse large B cell lymphoma (DLBCL). The acquisition of genetic lesions is an established oncogenic event in the molecular pathogenesis of DLBCL. The best characterized events are rearrangements of BCL6, BCL2 and MYC. BCL6 is a transcriptional repressor that protects germinal center B cells undergoing mutation and rearrangements from apoptosis, but its persistent expression may block exit from the germinal center and promote lymphomagenesis [61,62,63,64]. Normally, BCL6 is also a negative regulator of MYC and BCL2. This function is lost in DLBCL [65]. Oncogene BCL2 blocks apoptosis. Chromosomal translocation t(14;18) involving BCL2 locus, which is the hallmark of FCL, is also detected in DLBCL. MYC gene rearrangements similar to those seen in Burkitt lymphoma [translocation t(8;14)] occur in DLBCL together with copy number amplification [62].

Lymphocyte trafficking is also impaired by mutations in DLBCL, such as those of S1PR2, that keeps B cells in the germinal center, of GNA13, that encodes for a small G protein regulator of B cell motility, and of its partner GPCR named P2RY8, that confines B cells within the germinal center [66,67].

Additional events that lead to immune evasion include loss of function mutations that affect B2M, HLA-A,B and C, necessary for cytotoxic T cell recognition, and CD58, necessary for NK cell surveillance [68]. Moreover, PCDLBCL, LT has been substantially associated with increased expression of PD-L1 and PD-L2 resulting from translocations of PD-L1/PD-L2 which promote a permissive tumor immune microenvironment. The expression of surface ligands for the PD-1 immune checkpoint sets PCDLBCL, LT apart from PCFCL or PCMZL and is similar to that of primary central nervous system lymphomas and primary testicular lymphomas [69].

## 4. Antibody-Directed Therapies for the Treatment of Lymphoma

### 4.1. Investigative Approaches for CTCL

In recent decades, the understanding of the pathogenetic mechanisms in CTCL led to the identification of different candidate antigens for targeted mAbs that are summarized in Table 2 [49].

The optimal targeted approach against T cell antigens in the advanced stages of CTCL is still investigative and comparative prospective investigations are warranted to establish the preferential therapy [27]. Alemtuzumab is an anti CD52 humanized IgG1 mAb which was formerly available in onco-hematology for the treatment of chronic lymphocytic leukemia. CD52 is surface antigen that is broadly express across T and B cells, as well as monocytes [70]. Alemtuzumab was investigated in a phase II study in the setting of advanced MF and SS: response was reported in over half patients with better results in cases of erythroderma than in those of plaque or tumor involvement [4]. To improve the safety profile, which is burdened by infective and hematologic adverse events, low dose regimens of alemtuzumab have been studied with similar responses in erythroderma patients [71].

Mogamulizumab is a humanized mAb against CCR4, which is typically overexpressed on the surface of malignant T cells. This mAb is defucosylated to enhance antibody-dependent cellular cytotoxicity. Mogamulizumab has emerged as the standard of care for previously treated CTCL patients according to results of a phase III trial [5]. Results showed overall response rates in MF and in SS of 21% and 37%, respectively. The drug showed higher clinical benefit in the blood compartments than for cutaneous and lymph node involvement. Additionally, mogamulizumab is able to deplete T regulatory cells, which also express CCR4, resulting in an increased antitumor immune function that may be related to the risk of abnormal immune reactions, such as graft-versus-host disease observed in hematopoietic cell transplant recipients receiving this drug [72]. This evidence also suggests that mogamulizumab could be used in combination with other cancer immunotherapies, such as PD-1 checkpoint inhibitors [73]. Finally, the CCR4 antigen could be a promising target for the development of future ADCs to enhance efficacy on the skin compared to that achievable with current mAbs.

Brentuximab vedotin (BV) is an ADC that targets CD30-positive cancer cells. This ADC consists of an anti-CD30 chimeric monoclonal IgG1 antibody conjugated to a microtubule inhibitor (monomethyl auristatin E, MMAE) by a cleavable linker. After internalization, the ADC undergoes proteolytic cleavage in lysosomes; MMAE is then released from the complex and inhibits tubulin polymerization, leading to programmed cell death [74].

CD30 belongs to the tumor necrosis factor receptor family and is involved in T lymphocyte immune response and regulation processes. This receptor is universally expressed in classical Hodgkin lymphoma and anaplastic large cell lymphoma; it is variably expressed in PCALCL and LyP, but also in B cell lymphomas such as DLBCL, and in other CTCLs such as MF. However, it must be underlined that a clear and universally accepted cut-off level for assessing CD30-positivity in CTCLs has not been defined yet [75,76].

BV is now indicated for the treatment of adult patients with CD30+ CTCLs after at least one prior systemic therapy. The drug has proven to be effective in several clinical trials: in a phase II prospective multi-institution study, BV was administered to 32 patients with MF or SS (stages IB through IVB), with variable CD30 expression; all patients underwent at least one systemic therapy failure. The objective response rate was around 70%, demonstrating significant efficacy. Higher expression of CD30 (>5%) was associated with a better response to the drug when compared to lower expression (<5%) [77]. Furthermore, in an open-label single-center phase II trial, BV efficacy was assessed in 48 patients with primary cutaneous CD30+ lymphomas, including LyP, PCALCL, and transformed MF. The objective response rate was around 73% (100% in patients with LyP and PCALCL; 54% in patients with MF). Moreover, in patients with MF higher expression of CD30 (>50%) was associated with a better response to the drug if compared to low (<10%) or medium expression (10 to 50%) [78]. In the international open-label randomized phase III multicenter ALCANZA trial, BV was associated with a more durable response if compared to methotrexate or bexarotene in 128 pretreated patients with CD30+ MF or PCALCL. In particular, in patients with MF, the objective response rate lasting four months was 50% in the BV group compared with 10% in the bexarotene/methotrexate arm; moreover, in patients with PCALCL the objective response rate lasting four months was 75% in the BV group compared with 20% in the bexarotene/methotrexate arm [79]. The final analysis from the ALCANZA trial, performed after a median follow-up of 45.9 months, confirmed results of the primary analysis, highlighting superiority of BV over bexarotene/methotrexate [80].

Overall, BV displays a favorable safety profile. The most common adverse event reported is peripheral neuropathy: nerve damage does not seem to be a CD30-mediated mechanism but rather the effect of MMAE-mediated inhibition of microtubule-dependent axonal transport. Other relatively common adverse events are fatigue, nausea, upper respiratory tract infections, and neutropenia [74,81,82]. Despite the overall efficacy and safety profile, resistance to BV is a recognized event and its underlying mechanisms have been characterized. Escape may be due to a downregulation of the target CD30, an altered intracellular accumulation of the cytotoxic agent MMAE, or an overexpression of transporters—P-glycoprotein, codified by multidrug resistance-1 (MDR1)—exporting cytotoxic agents out of the cell [83,84]. Competitive inhibition of MDR1 could restore sensitivity to BV in BV-resistant cell lines in vitro by increasing intracellular MMAE levels [85].

To overcome resistance or to improve response to BV two main strategies are being currently explored, including the combination of BV with other drugs and the identification of new potential target antigens.

As for combination strategies, Hasanali et al. have shown that histone deacetylase inhibition in patients with alemtuzumab-resistant T cell prolymphocytic leukemia leads to CD30 upregulation, allowing for subsequent successful treatment with BV [86]. This evidence gave rise to a multicenter phase I clinical trial assessing feasibility of BV combined with romidepsin (a histone deacetylase inhibitor) in patients with CTCL: preliminary results showed good tolerability and response to this combination strategy [87].

As for new potential target antigens, two promising molecular targets are inducible T cell costimulator (ICOS) and cell surface heat shock protein 70 (csHSP70). ICOS was found to be highly expressed in skin biopsies from patients with MF and SS by Amatore et al.; consequently, an ADC composed of a murine anti-ICOS mAb conjugated to MMAE and pyrrolobenzodiazepine was tested in a preclinical model of a CTCL xenograft: the ADC showed an in vitro and in vivo efficacy superior to BV [88]. csHSP70 was also found to be highly expressed in skin specimens from patients with CTCLs; an ADC composed of an anti-csHSP70 mAb linked to MMAE was then generated and tested: in vitro comparison with BV revealed similar efficacy in MF and SS cell lines [89].

### 4.2. Investigative Approaches for CBCL

The standard treatment armamentarium for B cell NHL generally consists of immune-chemotherapy regimens containing mAbs against a B cell antigen such as CD20. The latter is an established target in B cell NHL owing to its expression in over 90% of malignant B cells [90]. However, some cases of B cell lymphoma (BCL) may show primary or secondary resistance to these targeted agents, motivating the research for alternative approaches and drug development strategies to bridge an unmet clinical need in the treatment of aggressive subtypes of BCL, including those primarily involving the skin [91].

A novel approach in haemato-oncology is the development of bi-specific T cell engagers (BiTE), that form a bridge between CD3 on the surface of cytotoxic T cells and a tumor-associated antigen [92]: blinatumomab, binding CD3 and CD19 on B cells, is approved by the FDA and the European Medicine Agency (EMA) for B cell precursor relapsed/refractory acute lymphoblastic leukemia. Preliminary evidence may also support its efficacy in BCLs, such as DLBCL [93]. An attractive option for expanding the BiTE approach may be provided by re-engineering conventional mAbs that are already available [3].

The PD-1/PD-L1 pathway could be a potential investigative target due to the surface expression by PCDLBCL, LT [69]. A 2019 study provided a proof of concept supporting this approach: a case of refractory PCDLBCL, LT in an elderly subject was treated with PD-1 blocker pembrolizumab in combination with rituximab and lenalidomide [94].

A different attempt to further increase the effect of mAbs is provided by the development of antibodies conjugated with radioisotopes, considering that B cell NHL is a malignancy that is extremely sensitive to radiation. Radioimmunotherapy for the treatment of B cell NHL is currently approved by the FDA using 90 Y-ibritumomab tiuxetan (90Y-I) and 131 I-tositumomab [95,96]. While the latter agent is no longer available on the market, the former has been used for relapsed/refractory lymphomas expressing CD20 as well as for consolidation following standard induction therapy [97]. Considering that this antibody targets B cell-specific antigen CD20, future application to the management of PCDLBCL, LT is predictable. However, it must be considered that prior use of rituximab—as usually occurs in first-line regimens—may reduce the effect of radioimmunotherapy. This hypothesis is supported by results of a 2007 study in refractory/relapsed cases of DLBDCL, showing higher response to 90Y-I in those who did not undergo previous rituximab [98]. Therefore, the placement of radioimmunotherapy with 90Y-I in the treatment scenario of aggressive CBCL is still speculative. Considering the favorable safety profile demonstrated by previous trials, where the main adverse effect consisted in a transient myelosuppression reverting to baseline in 2–3 months [99], this approach could be a feasible first-line alternative in elderly patients and other subjects that are unfit for standard combination regimens. Moreover, a shorter course of chemotherapy may be achieved by adding 90Y-IT for consolidation, reducing dose-related toxicity of traditional agents [100]. Finally, in the setting of relapsed disease, radioimmunotherapy could be advisable for selected cases that are not candidates for more aggressive approaches.

## 5. Future Perspectives

A further improvement in the landscape of targeted therapies is represented by near-infrared photoimmunotherapy (NIR-PIT). This approach is based on a near-infrared photo-inducible molecule linked to a mAb targeting a specific tumor antigen, that is an antibody–photoabsorber conjugate (APC) that is activated by NIR light. This way, only APC-bound tumor cells that are exposed to NIR light are killed, limiting adjacent normal cells damage. Moreover, it has been observed that NIR-PIT promotes a potent immune response which leads to further tumor cell or tumor microenvironment killing [101,102].

Silic-Benussi et al. investigated NIR-PIT in MF by targeting the CLA, which is expressed in skin-resident effector memory T cells of MF. Specifically, MF cells were incubated with an anti-CLA mAb conjugated to the hydrophilic phthalocyanine IRdye 700DX^®^ (IR700) and then irradiated with NIR light. The authors observed a very modest killing effect when using anti-CLA mAb or light irradiation alone, while the combination of the two led to substantial increase in death in the MF cell lines, with no damage to CLA-negative cells [103].

Finally, similar results were obtained on murine models of BCL expressing CD20 using rituximab conjugated with IR700 [104]. These preliminary results provide the proof of concept for the future development of NIR-PIT-based approaches in the management of cutaneous lymphoma.

## 6. Conclusions

The management of advanced cutaneous lymphoma is currently a challenge in the field of dermato-oncology, due to limited treatment options and unsatisfactory responses. Moreover, targeted approaches are penalized by lack of recommendations regarding preferential drugs and combination regimens. In this context, antibody-directed therapies, which include a spectrum of available and investigative approaches, represent an attractive addition to overcome the limitations of the current treatment armamentarium. Finally, the aim of improving efficacy while maintaining an excellent safety profile may be achieved with the development of targeted agents that can be selectively activated in the skin.

## Figures and Tables

**Figure 1 antibodies-12-00021-f001:**
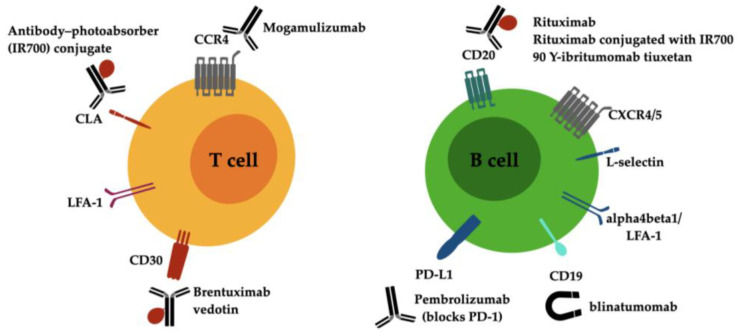
Drug targets on malignant lymphocytes of T and B cell cutaneous lymphoma and current antibody-directed treatment options. Abbreviations: CCR4 C-C chemokine receptor type 4, CD, cluster of differentiation; CLA, cutaneous lymphocyte antigen; CXCR4/5, C-X-C chemokine receptor type 4/5; LFA-1, lymphocyte function-associated antigen 1 (alphaLbeta2 integrin).

**Table 1 antibodies-12-00021-t001:** Classification of primary cutaneous lymphomas according to clinical behavior, following the WHO-EORTC classification and its 2018 revision.

Clinical Behavior	CTCL	CBCL
Indolent	MF PCALCL LyP Subcutaneous panniculitis-like T cell lymphoma Rare subtypes *	PCMZL PCFCL EBV+ mucocutaneous ulcer
Intermediate	PCALCL	PCDLBCL, LT PCDLBCL, other Primary cutaneous intravascular large B cell lymphoma
Aggressive	SS Rare subtypes **	PCDLBCL, LT

* Rare subtypes associated with an indolent course include primary cutaneous CD4+ small/medium T cell LPD and primary cutaneous acral CD8+ T cell lymphoma. ** Rare subtypes characterized by aggressive behavior include primary cutaneous NK/T cell lymphoma, nasal type; primary cutaneous aggressive epidermotropic CD8+ cytotoxic T cell lymphoma; primary cutaneous gamma-delta T cell lymphoma; primary cutaneous peripheral T cell lymphoma, unspecified. Abbreviations: CTCL, cutaneous T cell lymphoma; CBCL, cutaneous B cell lymphoma; EBV, Epstein–Barr virus; LPD, lymphoproliferative disorder; LyP, lymphomatoid papulosis; MF, mycosis fungoides and its variants, folliculotropic MF, pagetoid reticulosis, granulomatous slack skin; PCALCL, primary cutaneous anaplastic large cell lymphoma; PCDLBCL, LT, primary cutaneous diffuse large B cell lymphoma, leg type; PCFCL, primary cutaneous follicle center lymphoma; PCMZL, primary cutaneous marginal zone lymphoma; SS, Sezary syndrome.

**Table 2 antibodies-12-00021-t002:** Antigen targets and clinical trials to treat cutaneous T cell lymphoma with antibody–drug conjugates.

Pathway	Drug	Target	Evidence
MAPK	Sorafenib	Multikinase (B-RAF, VEGFR)	Pilot study
PI3K/Akt/mTOR	Duvelisib	PI3K-delta/gamma	Phase I
	Everolimus	mTOR	Pilot study
JAK/STAT	Ruxolitinib	JAK1/2	In vitro
	Tofacitinib	JAK1/2/3	In vitro
T cell	Nivolumab	PD-1	Phase I
	Pembrolizumab	PD-1	Phase II
	Ipilimumab	CTLA4	Case report
	Mogamulizumab	CCR4	Approved
	Alemtuzumab	CD52	(withdrawn)
	Resimmune	CD3	Phase II

Abbreviations: Akt, serine-threonine protein kinase B; B-RAF, B-Raf proto-oncogene, serine/threonine kinase; CCR, C-C chemokine receptor; CTLA4, cytotoxic T-lymphocyte-associated protein 4; JAK, Janus kinase; MAPK, mitogen-activated protein kinase; mTOR, mammalian target of rapamycin; PD-1, programmed cell death protein 1; PI3K, phosphatidylinositol-4,5-bisphosphate 3-kinase; STAT, signal transducer and activator of transcription; VEGFR, vascular endothelial growth factor receptor.

## Data Availability

No new data were created or analyzed in this study. Data sharing is not applicable to this article.

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
