# Peer review of "Cutaneous Lymphoma and Antibody-Directed Therapies"

_2073-4468, 2023, doi:10.3390/antib12010021_

Round 1
Reviewer 1 Report
This manuscript provides a very nice review of cutaneous lymphoma and antibody-related therapies in treating the disease. I really enjoyed reading it. Many thanks for the opportunity. Yet, some minor improvements as detailed below may further benefit our future readers.
After finishing reading the manuscript, I have to say my first impression is that the current title seems off. Section 4.1 did give a very nice summary about brentuximab vedotin, which is an anti-CD30 antibody-drug conjugate (ADC). Yet, the current manuscript did spend two out of five sections providing a detailed review of cutaneous lymphoma itself. Also, in addition to ADCs, the authors also discussed bispecific antibodies, antibody combinations, and radioimmunotherapy in Section 4.2, along with antibody–photo-absorber conjugates (APCs) in Section 5. I understand that the authors may consider radioimmunotherapy and APCs as subtypes of broadly defined ADCs. In this case, an elaboration on subtypes of ADCs would be highly recommended. Based on the thoughts above, a title like “a review of cutaneous lymphoma and antibody-directed therapies” or similar might be more suitable. Further, it seems that current Section 5 solely focuses on future perspectives, more particularly, APCs. Some conclusions in a further section are strongly suggested.
In addition, as the authors have already started doing in Section 4, detailed discussions about more antibodies under investigation in treating cutaneous lymphoma would be super beneficial to our readers. If possible, a table is suggested, listing antibodies and their targets under evaluation for treating cutaneous lymphoma. For example, the importance of CCR4 was discussed in Section 2.2 with respect to cutaneous T-cell lymphoma. Yet, the current manuscript remains silent on an anti-CCR4 antibody, mogamulizumab. See, Kim et al. Mogamulizumab versus vorinostat in previously treated cutaneous T-cell lymphoma (MAVORIC): an international, open-label, randomised, controlled phase 3 trial. Lancet Oncol. 2018 Sep;19(9):1192-1204. I understand that CCR4 may not have been explored as an ADC target. Still, an analysis of why or why not it can be a promising ADC target would be interesting. Additionally, by doing so, it can organically link the mechanism discussions in Sections 2 and 3 to the therapy reviews in Sections 4 and 5.
Further, the following minor changes are suggested for ease of reading and understanding.
(1) The current title recites “drug-antibody conjugates.” Yet, for consistency, it is suggested to amend it to “antibody-drug conjugates.”
(2) The term “soluble protein” in line 42 can be confusing. An alternative term, such as “intracellular protein,” seems more appropriate.
(3) Sezary syndrome (SS) and mycosis fungoides (MF) are believed two subtypes of cutaneous T cell lymphomas (CTCL). If this understanding is correct, does it make sense to add SS to the sentence in lines 68-70? In this way, we will have support for the statement of “90% of CTCL,” which includes 55% (MF), 5% (SS), and 30% (primary cutaneous CD30+ LPD).
(4) It seems an “and” is missing in line 138, while “int” in line 146 is believed meant to be “in.”
Author Response
Reviewer 1
This manuscript provides a very nice review of cutaneous lymphoma and antibody-related therapies in treating the disease. I really enjoyed reading it. Many thanks for the opportunity. Yet, some minor improvements as detailed below may further benefit our future readers.
After finishing reading the manuscript, I have to say my first impression is that the current title seems off. Section 4.1 did give a very nice summary about brentuximab vedotin, which is an anti-CD30 antibody-drug conjugate (ADC). Yet, the current manuscript did spend two out of five sections providing a detailed review of cutaneous lymphoma itself. Also, in addition to ADCs, the authors also discussed bispecific antibodies, antibody combinations, and radioimmunotherapy in Section 4.2, along with antibody–photo-absorber conjugates (APCs) in Section 5. I understand that the authors may consider radioimmunotherapy and APCs as subtypes of broadly defined ADCs. In this case, an elaboration on subtypes of ADCs would be highly recommended. Based on the thoughts above, a title like “a review of cutaneous lymphoma and antibody-directed therapies” or similar might be more suitable. Further, it seems that current Section 5 solely focuses on future perspectives, more particularly, APCs. Some conclusions in a further section are strongly suggested.
In addition, as the authors have already started doing in Section 4, detailed discussions about more antibodies under investigation in treating cutaneous lymphoma would be super beneficial to our readers. If possible, a table is suggested, listing antibodies and their targets under evaluation for treating cutaneous lymphoma. For example, the importance of CCR4 was discussed in Section 2.2 with respect to cutaneous T-cell lymphoma. Yet, the current manuscript remains silent on an anti-CCR4 antibody, mogamulizumab. See, Kim et al. Mogamulizumab versus vorinostat in previously treated cutaneous T-cell lymphoma (MAVORIC): an international, open-label, randomised, controlled phase 3 trial. Lancet Oncol. 2018 Sep;19(9):1192-1204. I understand that CCR4 may not have been explored as an ADC target. Still, an analysis of why or why not it can be a promising ADC target would be interesting. Additionally, by doing so, it can organically link the mechanism discussions in Sections 2 and 3 to the therapy reviews in Sections 4 and 5.
Dear Reviewer,
Thank you for your careful review of our paper: we have carefully considered each issue raised and modified the manuscript accordingly. Our responses are provided below.
We have elaborated on the types of ADC that are considered in the review in the abstract and in the introduction. Please see: “The authors reviewed antibody-directed therapies investigated in the setting of lymphoma: this term includes a broad spectrum of approaches, from antibody-drug conjugates such as brentuxi-mab vedotin, to bi-specific antibodies, antibody combinations, antibody-conjugated nanothera-peutics, radioimmunotherapy and, finally, photoimmunotherapy with specific anti-body-photoadsorber conjugates, as an attractive strategy in development for the future manage-ment of cutaneous lymphoma.” and “Then, evidence on the use of antibody-directed therapies in the setting of lymphoma is discussed: this broad definition includes ADCs as well as bi-specific T cell engagers, antibody combinations, antibody-functionalized nanotherapeutics, and radioimmuno-therapy. Finally, the development of antibodies conjugated with a photoadsorber mol-ecule is presented as an attractive strategy for the future management of cutaneous lymphoma, providing target specific binding with the additional advantage of limiting activation to the malignant tissue irradiated by light. This latter approach is particularly suitable in cutaneous oncology considering that the skin compartment is readily accessible to the effect of an external light source.”
We have added the section “6. Conclusions”; please see in the text: “The management of advanced cutaneous lymphoma is currently a challenge in the field of dermato-oncology, due to limited treatment options and unsatisfactory responses. Moreover, targeted approaches are penalized by lack of recommendations regarding preferential drugs and combination regimens. In this context, antibody-directed therapies – which include a spectrum of available and investigative approaches – represent an attractive addition to overcome the limitations of the current treatment armamentarium. Finally, the aim of improving efficacy while maintaining an excellent safety profile may be achieved with the development of targeted agents that can be selectively activated in the skin.”
We have added Table 2. Antigen targets and clinical trials to treat cutaneous T cell lymphoma with anti-body-drug conjugates.
The role of mogamulizumab in CTCL has been discussed and the relevant reference added. Please see in section 4.1: “Mogamulizumab is a humanized mAb against CCR4, which is typically overex-pressed on the surface of malignant T cells. This mAb is defucosylated to enhance an-tibody-dependent cellular cytotoxicity. Mogamulizumab has emerged as the standard of care for previously treated CTCL patients according to results of a phase III trial [5]. Results showed overall response rates in MF and in SS of 21% and 37%, respectively. The drug showed higher clinical benefit in the blood compartments than for cutaneous and lymph node involvement. Additionally, mogamulizumab is able to deplete T regulatory cells, which also express CCR4, resulting in an increased antitumor immune function that may be related to the risk of abnormal immune reactions, such as graft-versus-host disease observed in hematopoietic cell transplant recipients receiving this drug [72]. This evi-dence also suggests that mogamulizumab could be used in combination with other cancer immunotherapies, such as PD-1 checkpoint inhibitors [73]. Finally, the CCR4 antigen could be a promising target for the development of future ADCs to enhance efficacy on the skin compared to that achievable with current mAbs.”
Further, the following minor changes are suggested for ease of reading and understanding.
(1) The current title recites “drug-antibody conjugates.” Yet, for consistency, it is suggested to amend it to “antibody-drug conjugates.”
Thank you, we have modified the title according to your comments to better reflect the contents of our review: “Cutaneous lymphoma and antibody-directed therapies”.
(2) The term “soluble protein” in line 42 can be confusing. An alternative term, such as “intracellular protein,” seems more appropriate.
Thank you for your comments, to avoid a potential source of confusion we changed to “intracellular antigens”
(3) Sezary syndrome (SS) and mycosis fungoides (MF) are believed two subtypes of cutaneous T cell lymphomas (CTCL). If this understanding is correct, does it make sense to add SS to the sentence in lines 68-70? In this way, we will have support for the statement of “90% of CTCL,” which includes 55% (MF), 5% (SS), and 30% (primary cutaneous CD30+ LPD).
Thank you, we have clarified this sentence as follows: “Specifically, mycosis fungoides (MF), Sezary syndrome (SS), and primary cutaneous CD30+ lymphoproliferative disorders (LPD) account for 55, 5 and 30% of CTCL, respectively.”
Reviewer 2 Report
This review is comprehensive, well written describing facts and numbers up to date but sparse on discussion points, recommendations, suggestions, or the author's position in the interpretation of the data.
In the introduction, there is no mention on the potential reasons if any, as to why advanced cases of this class of tumors, (unless this is a general statement for treatment of all advanced stage neoplasias), respond poorly to standard treatment of care (chemo or antibody approaches). Therefore, it will be relevant to highlight the current thinking as to why the ADC approach could result in better outcomes (Brentuximab vedotin) in these conditions (advance stages) and malignancies (cutaneous lymphomas). On the other hand, the rational of proposing antibodies conjugated with a photoadsorber molecules, for skin tumors is self-evident.
Corrections:
#275 bexarotene in 128 pretreated patents
Please change “patents” to patients.
#325-326 An attractive option for expanding the BiTE approach may be provided by the re-engineering conventional mAbs that are already available
There is no mention of “BiTEs” in the abstract or the introduction. For consistency, please add a brief statement in either section.
Author Response
Reviewer 2
This review is comprehensive, well written describing facts and numbers up to date but sparse on discussion points, recommendations, suggestions, or the author's position in the interpretation of the data.
Dear Reviewer,
Thank you for your efforts on our paper and for your constructive comments. We have carefully responded to each issue raised and our responses are provided below point-by-point.
In the introduction, there is no mention on the potential reasons if any, as to why advanced cases of this class of tumors, (unless this is a general statement for treatment of all advanced stage neoplasias), respond poorly to standard treatment of care (chemo or antibody approaches). Therefore, it will be relevant to highlight the current thinking as to why the ADC approach could result in better outcomes (Brentuximab vedotin) in these conditions (advance stages) and malignancies (cutaneous lymphomas). On the other hand, the rational of proposing antibodies conjugated with a photoadsorber molecules, for skin tumors is self-evident.
Thank you for your constructive comment. We have added the following explanation to the text to clarify our statements and provide the necessary background to the management of cutaneous lymphoma. Please see in the introduction: “While regimens incorporating anti-CD20 mAbs may be considered the current standard of care in cutaneous B cell lymphoma (CBCL), the optimal management for advanced cutaneous T cell lymphoma (CTCL) has not been defined [16]. A sequential multidisci-plinary approach based on the disease stage and tailored individually is the preferred strategy and incorporates the sequential use or combination of biologic-response modi-fying drugs [17], histone deacetylase inhibitors [18,19], and extracorporeal photopheresis [20]. Moreover, escalated chemotherapy regimens may provide elevated rates of response but are burdened by adverse events – namely, infections and myelosuppression – and by a short duration of responses [21,22]. For these reasons, traditional chemotherapy is usually considered following multiples relapses or for the debulking of extensive visceral disease [21]. Relapses are also frequent following autologous stem cell transplant and only allogenic transplant may provide durable responses, mediated by a graft-versus-lymphoma reaction [23,24]. Finally, targeted agents require robust evidence supporting their use alone or in combination and additional novel approaches are ur-gently needed [25–27]. This review briefly summarizes current understanding of the pathophysiology of T and B cell lymphomagenesis with a focus on recognized molecular alterations that may provide investigative therapeutic targets. Then, evidence on the use of antibody-directed therapies in the setting of lymphoma is discussed: this broad defi-nition includes ADCs as well as bi-specific T cell engagers, antibody combinations, antibody-functionalized nanotherapeutics, and radioimmunotherapy. Finally, the development of antibodies conjugated with a pho-toadsorber molecule is presented as an attractive strategy for the future management of cutaneous lymphoma, providing target specific binding with the additional advantage of limiting activation to the malignant tissue irradiated by light. This latter approach is particularly suitable in cutaneous oncology considering that the skin compartment is readily accessible to the effect of an external light source.”
Corrections:
#275 bexarotene in 128 pretreated patents
Please change “patents” to patients.
We apologize for this spelling mistake. Please see the correct sentence: “bexarotene in 128 pretreated patients”
#325-326 An attractive option for expanding the BiTE approach may be provided by the re-engineering conventional mAbs that are already available
There is no mention of “BiTEs” in the abstract or the introduction. For consistency, please add a brief statement in either section.
Thank you, we have mentioned the types of antibody-directed therapies that have been considered, in the introduction (please see comment above) and in the abstract. Please see in the abstract: “The authors reviewed antibody-directed therapies investigated in the setting of lymphoma: this term includes a broad spectrum of approaches, from antibody-drug conjugates such as brentuxi-mab vedotin, to bi-specific antibodies, antibody combinations, antibody-conjugated nanothera-peutics, radioimmunotherapy and, finally, photoimmunotherapy with specific anti-body-photoadsorber conjugates, as an attractive strategy in development for the future manage-ment of cutaneous lymphoma.”
Reviewer 3 Report
A. Sernicola and co-workers have submitted an informative review of the effectiveness of antibody-drug conjugates (ADCs) in setting of lymphoma as an attractive strategy for the future management of cutaneous lymphoma. Definitely, the topic is of high interest for the readership because ADCs are highly noticed in the clinic and market. Also, a logical classification for T cell and B cell lymphoma is given by the authors which is informative. I would recommend acceptance of this manuscript for publication by Antibodies after giving a “minor revision” by the authors. My comments are as below;
1. In the title, “Drug-antibody conjugates” should be converted to “Antibody-drug conjugates” because this generation of the advanced anticancer medications is known as “ADCs”.
2. In the introduction section, authors stated “However, mAbs alone do not often provide sufficient cytotoxic effects, resulting in limited response 34 when employed as monotherapy”, but they did not expressed the reasons. Please shortly discuss.
3. Authors stated “MAbs usually belong to the IgG subtypes and are humanized or chimeric in order to minimize their immunogenicity”. I would suggest authors to briefly review different types of antibodies before this sentence because the broad readership of this journal may include researchers in various fields except medicine.
4. For the introduction section, I strongly recommend authors to discuss “antibody-conjugated nanomedicine”, too. Recently, attentions to this generation of antibody-functionalized nano-therapeutics has increased because many drawbacks (e.g. low stability in the physiological environment) is addressed through using conjugated nano-systems. In this regard, the authors should cite the pioneering work in the field, as follows: Commun Biol 5, 995 (2022), https://doi.org/10.1038/s42003-022-03966-w
5. The whole manuscript lacks informative tables and eye-catching images! Please add.
6. Pharmacokinetics of ADCs for the treatment of cutaneous lymphoma should be discussed in a separate section. In this section, authors should review highlights and developments in recent decade.
Author Response
Reviewer 3
- Sernicola and co-workers have submitted an informative review of the effectiveness of antibody-drug conjugates (ADCs) in setting of lymphoma as an attractive strategy for the future management of cutaneous lymphoma. Definitely, the topic is of high interest for the readership because ADCs are highly noticed in the clinic and market. Also, a logical classification for T cell and B cell lymphoma is given by the authors which is informative. I would recommend acceptance of this manuscript for publication by Antibodies after giving a “minor revision” by the authors. My comments are as below;
Dear Reviewer,
Thank you for reviewing our paper and for your constructive comments. We have carefully considered your comments and our responses are provided below point-by-point.
- In the title, “Drug-antibody conjugates” should be converted to “Antibody-drug conjugates” because this generation of the advanced anticancer medications is known as “ADCs”.
Thank you, we have modified the title according to your comments and to those of Reviewer 1 to better reflect the contents of our review: “Cutaneous lymphoma and antibody-directed therapies”.
- In the introduction section, authors stated “However, mAbs alone do not often provide sufficient cytotoxic effects, resulting in limited response 34 when employed as monotherapy”, but they did not expressed the reasons. Please shortly discuss.
Thank you for your constructive comment, we have added an explanatory sentence to clarify our statement. Please see: “This holds true for the mAbs currently available in the setting of cutaneous lymphoma which achieve response in around half of patients, with ample variations according to stage and differential involvement of the skin and blood compartments [4,5].”
- Authors stated “MAbs usually belong to the IgG subtypes and are humanized or chimeric in order to minimize their immunogenicity”. I would suggest authors to briefly review different types of antibodies before this sentence because the broad readership of this journal may include researchers in various fields except medicine.
Thank you for your suggestion, we have added a brief statement to provide the necessary context on the types of monoclonal antibodies. Please see in the text: “MAbs were first developed in murine models; in order to reduce their immunogenicity when used in humans, chimeric antibodies and humanized antibodies were developed, by combining sequences of the murine variable domain with human constant domain and by inserting the murine hypervariable regions into a human antibody, respectively. Finally, the development of additional techniques allowed fully human monoclonal antibodies to be derived [6,7]. MAbs used to develop ADCs usually belong to the IgG subtypes and are humanized or chimeric to minimize their immunogenicity;”
- For the introduction section, I strongly recommend authors to discuss “antibody-conjugated nanomedicine”, too. Recently, attentions to this generation of antibody-functionalized nano-therapeutics has increased because many drawbacks (e.g. low stability in the physiological environment) is addressed through using conjugated nano-systems. In this regard, the authors should cite the pioneering work in the field, as follows: Commun Biol 5, 995 (2022), https://doi.org/10.1038/s42003-022-03966-w; Heliyon, 2022, 8, 6, e09577 https://doi.org/10.1016/j.heliyon.2022.e09577; J Nanobiotechnol 19, 239 (2021) https://doi.org/10.1186/s12951-021-00982-6; Life Sciences, 2020, 240, 117099, https://doi.org/10.1016/j.lfs.2019.117099.
Thank you for your constructive comment. We have added a paragraph presenting this interesting and novel topic to the introduction, together with the relevant citations. Please see: “An additional advancement in the field of antibody-directed therapies consists in the development of antibody-functionalized nano-therapeutics. The generation of nanobi-oconjugates, that is drug-containing nanocarriers, is an attractive option for selective cancer therapy [12,13]. A further level of selectivity is provided by making these nanostructures responsive to specific stimuli, including externally applied irradiation [14,15]. Antibody-conjugated nanomedicine could combine the target-specific binding of mAbs with enhanced delivery and on-demand release of drugs, achieving high efficacy and excellent safety in future oncologic therapeutics.”
- The whole manuscript lacks informative tables and eye-catching images! Please add.
Thank you for your constructive comment. Please see in the revised manuscript:
- Table 1. Classification of primary cutaneous lymphomas according to clinical behavior, follow-ing the WHO-EORTC classification and its 2018 revision.
- Figure 1. Drug targets on malignant lymphocytes of T and B cell cutaneous lymphoma and cur-rent antibody-directed treatment options.
- Table 2. Antigen targets and clinical trials to treat cutaneous T cell lymphoma with anti-body-drug conjugates.
- Pharmacokinetics of ADCs for the treatment of cutaneous lymphoma should be discussed in a separate section. In this section, authors should review highlights and developments in recent decade.
Thank you for your constructive comment. Although an extensive discussion of the pharmacokinetics of ADC is considered beyond the scope of the current paper, we have added a relevant statement according to your suggestions. Please see in the text: “The pharmacokinetics of mAbs mainly involves nonlinear and saturable clearance which is mediated by binding to the target – a cell surface receptor of peripheral blood mononuclear cells –; additionally, at higher doses non saturable proteolytic pathways are able to degrade IgG antibodies similarly to endogenous IgG [6].”
Reviewer 4 Report
The review article provides a nice summary of cutaneous lymphomas, its types and pathogenesis and treatment options looking into the future. While the authors present valuable information, I have few points that I think could elevate the article.
The review article is very text heavy and I recommend adding figures/tables to give a change of pace to the reader who otherwise would find the article monotonous. For instance, the classification of lymphomas can be easily summarized as a tree diagram or a table. Also, the target audience needs to be given a more careful thought. If the review article is meant for a broader audience that may not be familiar with lymphomas, the authors need to provide more background, build connections between concepts, make the language more lucid, and educate the reader. Furthermore, more processing of the information/data needs to be done to give insights into the current status of the field and future steps. As of now, at many places the article merely tends to summarize information from published articles. The therapy section needs to build a more cohesive story starting with what has been used and progressing to the desirable future state. Lastly, the discussion/conclusion section does not summarize/discuss the information presented at all. It needs to be more lengthy and comprehensive where the authors can present their in-depth analysis.
Few other minor comments:
1) Curious why the title uses drug-antibody conjugates? the convention in the field is antibody-drug conjugates which is what is used in the rest of the manuscript.
2) Reference needed for line 39-40 where ADCs are introduced.
3) Reference needed for line 42 for soluble antigens.
Author Response
Reviewer 4
The review article provides a nice summary of cutaneous lymphomas, its types and pathogenesis and treatment options looking into the future. While the authors present valuable information, I have few points that I think could elevate the article.
The review article is very text heavy and I recommend adding figures/tables to give a change of pace to the reader who otherwise would find the article monotonous. For instance, the classification of lymphomas can be easily summarized as a tree diagram or a table. Also, the target audience needs to be given a more careful thought. If the review article is meant for a broader audience that may not be familiar with lymphomas, the authors need to provide more background, build connections between concepts, make the language more lucid, and educate the reader. Furthermore, more processing of the information/data needs to be done to give insights into the current status of the field and future steps. As of now, at many places the article merely tends to summarize information from published articles. The therapy section needs to build a more cohesive story starting with what has been used and progressing to the desirable future state. Lastly, the discussion/conclusion section does not summarize/discuss the information presented at all. It needs to be more lengthy and comprehensive where the authors can present their in-depth analysis.
Dear Reviewer,
Thank you for efforts on our manuscript and for the constructive comments. According to your suggestions, we have added relevant tables and figures. Please see in the revised manuscript:
- Table 1. Classification of primary cutaneous lymphomas according to clinical behavior, follow-ing the WHO-EORTC classification and its 2018 revision.
- Figure 1. Drug targets on malignant lymphocytes of T and B cell cutaneous lymphoma and cur-rent antibody-directed treatment options.
- Table 2. Antigen targets and clinical trials to treat cutaneous T cell lymphoma with anti-body-drug conjugates.
Additionally, a table of contents and list of abbreviations have been added to improve overall cohesion.
To provide the necessary background on the management of cutaneous lymphoma we have made extensive additions to the introduction section (please see 1. Introduction).
Finally, to offer the necessary insight on the topic, we have elaborated on the available and investigative antibody-directed therapeutics for CTCL (please see section 4.1 and Table 2).
Few other minor comments:
1) Curious why the title uses drug-antibody conjugates? the convention in the field is antibody-drug conjugates which is what is used in the rest of the manuscript.
Thank you, we have modified the title according to your comments and to those of Reviewer 1 to better reflect the contents of our review: “Cutaneous lymphoma and antibody-directed therapies”.
2) Reference needed for line 39-40 where ADCs are introduced.
Thank you, the relevant reference has been added. Please see reference [6]: Hoffmann, R.M.; Coumbe, B.G.T.; Josephs, D.H.; Mele, S.; Ilieva, K.M.; Cheung, A.; Tutt, A.N.; Spicer, J.F.; Thurston, D.E.; Crescioli, S.; et al. Antibody Structure and Engineering Considerations for the Design and Function of Antibody Drug Conjugates (ADCs). Oncoimmunology 2018, 7, e1395127, doi:10.1080/2162402X.2017.1395127.
3) Reference needed for line 42 for soluble antigens.
Thank you for your observation: the missing reference has been added. Please see reference [7]: Esnault, C.; Schrama, D.; Houben, R.; Guyétant, S.; Desgranges, A.; Martin, C.; Berthon, P.; Viaud-Massuard, M.-C.; Touzé, A.; Kervarrec, T.; et al. Antibody-Drug Conjugates as an Emerging Therapy in Oncodermatology. Cancers (Basel) 2022, 14, 778, doi:10.3390/cancers14030778.
Reviewer 5 Report
This is a very interesting review that highlights the use of antibody-photoadsober conjugates as an alternative treatment of cutaneous lymphoma. The authors discussed in detail about cutaneous T and B cell lymphoma and mechanism underlying the disease. The review appears to have been thoroughly discussed carefully about Brentuximab vedotin (BV). However, there are MANY PLACES in the review in which the presentation must be improved as noted below.
There should be a table of contents and abbreviations used in review is encouraged and a figure depicting B and T cell cutaneous lymphoma and drug targets and current treatment options and also a table showing ongoing clinical trials to treat cutaneous lymphoma with drug antibody conjugates.
1. Line 95: continuation is missing after “microenvironment”
2. Line 145: reference is missing for “An increased MAPK signaling due to gain of function 144 mutations in B-RAF and ERK-1 has been demonstrated in subjects with MF” study.
3. Line 154: List some key proteins of signaling pathways that stimulate malignant T cells.
4. Line 165: For “sorted into three main subtypes” Mention as 1,2 and 3 correspondingly.
5. Line 166: Reference for “2005 WHO-EORTC classification”
6. Line 169: What is “t”
7. Line 184: Remove “– such as that “
8. Line 185: Should be “and” instead of “- or to “
9. Line 229: What is “T”
10. Line 229: should be 14,18
11. Line 231: What is “t”
12. Line 231: should be 8,14
13. Line 235: should be GPCR named P2RY8
14. Line 248: elaborate with list of antigens and their target antibodies
15. Line 247: Should be Brentuximab vedotin (BV) is an ADC “targets CD30 positive cancer cells”
16. Line 266: remove moreover and should be “drug when compared.”
Author Response
Reviewer 5
This is a very interesting review that highlights the use of antibody-photoadsober conjugates as an alternative treatment of cutaneous lymphoma. The authors discussed in detail about cutaneous T and B cell lymphoma and mechanism underlying the disease. The review appears to have been thoroughly discussed carefully about Brentuximab vedotin (BV). However, there are MANY PLACES in the review in which the presentation must be improved as noted below.
Dear Reviewer,
Thank you for your careful review of our paper and for the encouraging comment. Please find our responses below in a point-by-point manner.
There should be a table of contents and abbreviations used in review is encouraged and a figure depicting B and T cell cutaneous lymphoma and drug targets and current treatment options and also a table showing ongoing clinical trials to treat cutaneous lymphoma with drug antibody conjugates.
Thank you for your constructive comment. We have added a table of contents and list of abbreviations before the Introduction section, according to your suggestions.
Additionally, the following tables and figure have been added to the revised manuscript:
- Table 1. Classification of primary cutaneous lymphomas according to clinical behavior, follow-ing the WHO-EORTC classification and its 2018 revision.
- Figure 1. Drug targets on malignant lymphocytes of T and B cell cutaneous lymphoma and cur-rent antibody-directed treatment options.
- Table 2. Antigen targets and clinical trials to treat cutaneous T cell lymphoma with anti-body-drug conjugates.
- Line 95: continuation is missing after “microenvironment”
Thank you, this sentence has been rewritten to improve clarity and avoid a potential source of confusion. Please see in the text: “Adhesion between lymphocytes and endothelial cells is mediated by molecules expressed on the cell surface and promoted by the secretion of chemokines which occurs in the context of local inflammation.”
- Line 145: reference is missing for “An increased MAPK signaling due to gain of function 144 mutations in B-RAF and ERK-1 has been demonstrated in subjects with MF” study.
Thank you, the missing reference has been added. Please see 46. Da Silva Almeida, A.C.; Abate, F.; Khiabanian, H.; Martinez-Escala, E.; Guitart, J.; Tensen, C.P.; Vermeer, M.H.; Rabadan, R.; Ferrando, A.; Palomero, T. The Mutational Landscape of Cutaneous T Cell Lymphoma and Sézary Syndrome. Nat Genet 2015, 47, 1465–1470, doi:10.1038/ng.3442.
- Line 154: List some key proteins of signaling pathways that stimulate malignant T cells.
Thank you for your suggestion. We have expanded this concept in Figure 1 and Table 2. The sentence has also been rephrased: “The currently available antibody-directed treatments for cutaneous lymphoma are summarized in Figure 1. Novel investigative strategies for drug discovery in CTCL will be provided by targeting the key proteins of signaling pathways that stimulate malignant T cells and firing up anti-tumor immune responses in the CTCL microenvironment [49].”
- Line 165: For “sorted into three main subtypes” Mention as 1,2 and 3 correspondingly.
Thank you, the three entities have been numbered as follows: “1) primary cutaneous follicle center lymphoma (PCFCL) …; 2) primary cutaneous marginal zone lymphoma (PCMZL) …; 3) primary cutaneous diffuse large B cell lymphoma, leg type (PCDLBCL, LT) is an intermediate/aggressive type that accounts for 20% of primary CBCL.”
- Line 166: Reference for “2005 WHO-EORTC classification”
Thank you, references have been replaced in the correct position in the text: “the 2005 WHO-EORTC classification for cutaneous lymphomas [28] and its 2018 revision [29]
- Willemze, R.; Jaffe, E.S.; Burg, G.; Cerroni, L.; Berti, E.; Swerdlow, S.H.; Ralfkiaer, E.; Chimenti, S.; Diaz-Perez, J.L.; Duncan, L.M.; et al. WHO-EORTC Classification for Cutaneous Lymphomas. Blood 2005, 105, 3768–3785, doi:10.1182/blood-2004-09-3502.
- Willemze, R.; Cerroni, L.; Kempf, W.; Berti, E.; Facchetti, F.; Swerdlow, S.H.; Jaffe, E.S. The 2018 Update of the WHO-EORTC Classification for Primary Cutaneous Lymphomas. Blood 2019, 133, 1703–1714, doi:10.1182/blood-2018-11-881268.”
- Line 169: What is “t”
Thank you, to improve clarity, the sentence has been rephrased as follows: “the hallmark chromosomal translocation t(14;18) the hallmark chromosomal translocation t(14;18)”
- Line 184: Remove “– such as that “
Thank you, the unnecessary expression has been removed “persistent inflammation consequent to chronic infection – caused by Borrelia burgdorferi – and autoimmunity”
- Line 185: Should be “and” instead of “- or to “
Thank you, this has been replaced according to your suggestion (please see above).
- Line 229: What is “T”
Thank you, the sentence has been reworded as follows: “Chromosomal translocation t(14;18)…”
- Line 229: should be 14,18
Thank you for your suggestion. The conventional typography uses the semicolon between the two chromosomes that are involved in the translocation (t). Please see, for example: Galteland, E., Sivertsen, E., Svendsrud, D. et al. Translocation t(14;18) and gain of chromosome 18/BCL2: effects on BCL2 expression and apoptosis in B-cell non-Hodgkin's lymphomas. Leukemia 19, 2313–2323 (2005). https://doi.org/10.1038/sj.leu.2403954
- Line 231: What is “t”
Thank you, the word “translocation” was added to improve clarity: “translocation t(8;14)”
- Line 231: should be 8,14
Thank you for your suggestion. The conventional typography uses the semicolon between the two chromosomes that are involved in the reciprocal chromosome translocation (t). For example, consider: Haluska, F., Finver, S., Tsujimoto, Y. et al. The t(8; 14) chromosomal translocation occurring in B-cell malignancies results from mistakes in V–D–J joining. Nature 324, 158–161 (1986). https://doi.org/10.1038/324158a0
- Line 235: should be GPCR named P2RY8
Thank you, we have corrected this mistake: “GPCR named P2RY8”
- Line 248: elaborate with list of antigens and their target antibodies
- Line 247: Should be Brentuximab vedotin (BV) is an ADC “targets CD30 positive cancer cells”
Thank you, we have corrected this sentence according to your suggestions: “Brentuximab vedotin (BV) is an ADC that targets CD30-positive cancer cells.”
- Line 266: remove moreover and should be “drug when compared.”
Thank you for your constructive comment, the sentence was corrected as follows: “Higher expression of CD30 (>5%) was associated with a better response to the drug when compared to lower expression (<5%)”